civil engineering

crack propagation, elastic–plastic, meshfree method, local radial point interpolation method, eXtended finite-element method

**Author for correspondence:**
Nengxiong Xu
e-mail: xunengxiong@cugb.edu.cn

# Comparative modelling of crack propagation in elastic–plastic materials using the meshfree local radial basis point interpolation method and eXtended finite-element method

Yazhe Li[1,2], Nengxiong Xu[1], Jinzhi Tu[1] and Gang Mei[1]

[1]School of Engineering and Technology, China University of Geosciences (Beijing), 100083 Beijing, People's Republic of China
[2]Henan Provincial Communications Planning and Design Institute Co., Ltd, 450000 Zhengzhou, People's Republic of China

YL, 0000-0002-2506-7688; NX, 0000-0003-2576-1546;
GM, 0000-0003-0026-5423

The modelling and understanding of crack propagation for elastic–plastic materials is critical in various engineering applications, such as safety analysis of concrete structures and stability analysis of rock slopes. In this paper, the local radial basis point interpolation method (LRPIM) combined with elastic–plastic theory and fracture mechanics is employed to analyse crack propagation in elastic–plastic materials. Crack propagation in elastic–plastic materials is compared using the LRPIM and eXtended finite-element method (XFEM). The comparative investigation indicates that: (i) the LRPIM results are close to the model test results, which indicates that it is feasible for analysing the crack growth of elastic–plastic materials; (ii) compared with the LRPIM, the XFEM results are closer to the experimental results, showing that the XFEM has higher accuracy and computational efficiency; and (iii) compared with the XFEM, when the LRPIM method is used to analyse crack propagation, the propagation path is not smooth enough, which can be explained as the crack tip stress and strain not being accurate enough and still needing further improvement.

# 1. Introduction

Fracture is a manifestation of material failure. Griffith and Orowan have done early work and made a breakthrough in brittle fracture theory [1]. As a special material, rock masses have the characteristics of both a continuous and discontinuous medium. On the one hand, under the action of force, a rock mass will undergo elastic–plastic deformation, and there are quite a number of constitutive models to describe the deformation characteristics. On the other hand, there are many structural planes in a rock mass, including splitting, joints and faults. These structural planes have different mechanical characteristics that cause the rock mass to have discontinuous deformation and strength characteristics. How to accurately calculate the stress–strain state of a rock with structural planes has become a hot topic.

Scholars have put forward various theories and calculation methods to study the discontinuous characteristics of rock masses. The main numerical methods for simulating fracture propagation are the finite-element method (FEM) [2], eXtended finite-element method (XFEM) [3] and meshless method [4]. When the FEM is used to study crack propagation, it is necessary to re-mesh the elements, which significantly increases the workload and leads to a low calculation accuracy and efficiency. These shortcomings can be avoided by the XFEM proposed by Belytschko and Moës [3–5]. Literature [6] investigates the crack propagation behaviour of asphalt materials through the use of damage models. The XFEM realizes accurate crack descriptions rationally and has become a research hotspot among numerical methods for crack problems [7].

Compared with the FEM, the meshless method is based on field nodes that can eliminate or partially eliminate the difficulties caused by meshing. The interpolation (approximation) function of the meshless method does not depend on elements, thus, it has the advantages of an easy higher order approximation formation and realization of local node encryption, bringing convenience to crack numerical analysis and making it more advantageous than the XFEM.

The element free Galerkin method (EFGM) uses the moving least squares method to construct the shape function, and obtains the governing equation from the weak form of the energy functional, thus obtaining the numerical solution of the partial differential equation. Belytschko et al. [8] first used the EFGM to analyse the fracture model and further applied it to the calculation of fracture expansion. Krysl & Belytschko [9] further extended the EFGM to fracture propagation calculations in a three-dimensional model. Belytschko et al. [10] applied the meshless method to simulate dynamic crack propagation in concrete structures, and they simulated the crack propagation direction and velocity in concrete structures very well. In [11], a meshless model was established based on the micro-image data, and the effects of aggregate, binder and voids on crack propagation were studied by calculation. Bordas et al. [12] presented a three-dimensional, extrinsically enriched meshfree method for the initiation, branching, growth and coalescence of an arbitrary number of cracks in nonlinear solids, including large deformation, for statics and dynamics. Zhuang et al. [13,14] applied the level set theory to calculate fracture extension in two-dimensional and three-dimensional models. The gap expansion calculation method, a three-dimensional meshfree method for modelling arbitrary crack initiation and growth in reinforced concrete structures, is presented in the literature [15].

All the above applications use the meshless EFGM, but this method requires the background grid when calculating the integration and is not a true meshless method. The local radial basis point interpolation method (LRPIM) [16] is a local weak computing method that does not require the background grid and is a real meshless method. The LRPIM is used to solve the two-dimensional time-dependent Maxwell equations in the literature [17]. The paper [18] presents a novel adaptive surrogate model for the probabilistic analysis of an aero-engine turbine disc by integrating the LRPIM and directional sampling technique. Free vibration analysis of composite laminates with delaminations is performed based on a three-dimensional semi-analytical model established by introducing the LRPIM into a Hamilton system in the literature [19].

Because the shape functions and their derivatives are discontinuous across the crack face and they have a severe variation near the crack tip, it is usually difficult to obtain an accurate solution at the crack tip. Therefore, many scholars obtain the exact solution [20–22] by strengthening the shape function of the crack tip, and some scholars obtain the exact solution [23] with the smooth point interpolation method. The LRPIM is used to analyse the problem of three-dimensional cracking in the literature [24,25].

Little research have been done on crack problems with LRPIM. Three-dimensional crack propagation of elastic–plastic materials with LRPIM have not yet, to our knowledge, been studied. In this paper, the polynomial basis function [26] and compactly supported radial basis function [27] are combined to construct the radial basis point interpolation shape function in a meshless method. According to the elastic–plastic theory and fracture mechanics, a three-dimensional elastic–plastic material crack propagation calculation is established. The algorithm and XFEM are used to solve the three-point bending beam to discuss the correctness and stability of the solution.

This paper is organized as follows: §2 briefly introduces the background of LRPIM and XFEM; §3 mainly concentrates on the methods of modelling crack propagation with LRPIM under elastoplastic conditions; §4 discusses the results; and finally, §6 draws several conclusions.

# 2. Background: the local radial basis point interpolation method and eXtended finite-element method

In this section, we briefly introduce the basic principles of the LRPIM and XFEM.

## 2.1. The meshfree local radial basis point interpolation method

At present, many meshless methods have been proposed by scholars. Their main differences lie in the approximation method and form of the system equation. The approximation method of LRPIM is radial basis point interpolation, and the system equation is obtained using the local weighted residual method:

$$\int_{\Gamma_q} \widehat{W_I} \sigma_{ij} n_j \, \mathrm{d}\Gamma - \int_{\Omega_q} \left[ \widehat{W_{I,j}} \sigma_{ij} - \widehat{W_I} b_i \right] \mathrm{d}\Omega = 0. \tag{2.1}$$

Unlike the global weak form, this method does not require the background mesh for integration and is a true meshless method. In this paper, the combination of a polynomial basis function [26] and compactly supported radial basis function [27] in a meshless method is used to construct the radial basis point interpolation shape function. The expression of its approximate function is

$$u(x) = \sum_{i=1}^{n} R_i(x) a_i + \sum_{j=1}^{m} p_j(x) b_j = R^{\mathrm{T}}(x) a + P^{\mathrm{T}}(x) b. \tag{2.2}$$

$R_i(x)$ is the radial basis function, $n$ is the number of radial basis functions, $p_j(x)$ is the monomial expression in space coordinates $x^{\mathrm{T}} = [x, y, z,]$, $m$ is the number of polynomial basis functions, and $a_i$ and $b_j$ are the undetermined coefficients. Generally, to ensure better stability, $n$ is larger than $m$.

The basis $p_j(x)$ of the polynomial can be determined using a Pascal triangle. In this paper, the polynomial in the three-dimensional case uses $m = 10$:

$$p_{j(x)} = (1 + x + y + z + xy + xz + yz + x^2 + y^2 + z^2)_j. \tag{2.3}$$

The expression of the compactly supported radial basis functions $R_i(x)$ is as follows:

$$R_i \left( 1 - \frac{r}{\delta} \right)^5 \left( 8 + 40 \frac{r}{\delta} + 48 \frac{r^2}{\delta^2} + 25 \frac{r^3}{\delta^3} + 5 \frac{r^4}{\delta^4} \right). \tag{2.4}$$

$\delta$ is the radius of the local support domain; $r$ is the distance between the integral point $x$ and the field node $x_i$ in the local support domain:

$$r = \sqrt{(x - x_i)^2 + (y - y_i)^2 + (z - z_i)^2}. \tag{2.5}$$

Considering the solid mechanics problem defined in domain $\Omega$, the local weighted residual method is used for node $I$ to satisfy the governing equation and the local weak equation of the node is obtained. The form of the locally weighted residuals is defined on the local integral domain $\Omega_q$ and the corresponding boundary $\Gamma_q$ in the following form:

$$\int_{\Omega_q} \widehat{W_I} (\sigma_{ij,j} + b_j) \mathrm{d}\Omega = 0. \tag{2.6}$$

$\widehat{W_I}$ is a weight function or test function centred on field node $I$. Equation (2.6) is applicable to all of the field nodes in the problem domain.

The discrete system equations in matrix form are deduced. The final system equations can be obtained by assembling all these equations based on the system of population numbers:

$$F_{3N*3N} U_{3N*1} = F_{3N*1}. \tag{2.7}$$

For cubic columns or cantilever beams with simple regular boundaries, the method of compression of the integration domain is adopted to solve this problem. For crack expansion, the relationship between nodes and Gauss integral points is constantly changing. Therefore, the visual criterion is used to deal with the range of the influence domain and the relationship between the field nodes and the Gauss

integral points. If the line between the field node and the Gauss integral point (either solid boundary or crack) intersects, it is considered that the node is 'masked' by the boundary and does not participate in calculating the Gauss integral point. After assembling the total rigid matrix, a penalty function is used to apply the displacement boundary conditions.

## 2.2. The eXtended finite-element method

Unit decomposition is the basis of XFEM. The basic idea of the decomposition method is that any function $\Psi(x)$ can be expressed as follows in the solving domain $\Omega$ [28]:

$$\Psi(x) = \sum_j N_j(x)\Phi(x). \tag{2.8}$$

$N_j(x)$ satisfies that: $\sum_j N_j(x) = 1$, which also satisfies the condition of function reconstruction [29], and $\phi(x)$ is an extension function. The parameter $q_j$ pair can be introduced to adjust the segment direction to reach the optimum approximation of $\Psi(x)$:

$$\Psi(x) = \sum_j N_j(x)q_j\Phi(x). \tag{2.9}$$

XFEM can describe complex unknown fields (such as discontinuous displacement fields) more accurately by adding extension terms on the basis of the standard field approximation. In XFEM, the finite-element approximation of the unknown field $u^h$ consists of two parts:

$$u^h = \sum_i N_i(x)u_i + \Psi(x), \tag{2.10}$$

where, $N_i(x)$ is the shape function of the standard FEM, $u_i$ is the degree of freedom of the standard node and the expansion $\Psi(x)$ is used to improve the characteristics of the unknown field. Using the unit decomposition attribute [30], the expression is

$$u^h = \sum_i N_i(x)u_i + \sum_j N_j(x)q_j\Phi(x). \tag{2.11}$$

The first term on the right is the standard finite-element approximation and the second term on the right is the extended term approximation based on unit decomposition, where $q_j$ is the node freedom of the new element. Compared with the traditional FEM, the most significant difference of XFEM is the introduction of redundant degrees of freedom at element nodes.

At present, the commercial software ABAQUS has a fracture analysis module [31]. To perform fracture analysis, the propagation functions usually include near-tip asymptotic functions to simulate the stress singularity near the crack tip and discontinuous functions to represent the displacement jump at the crack surface:

$$u = \sum_{I=1}^{N} N_I(x)\left[u_I + H(x)a_I + \sum_{\alpha}^{4} F_{\alpha}(x)b_I^{\alpha}\right]. \tag{2.12}$$

Among them, $N_I(x)$ is the commonly used nodal displacement shape function. The first term on the right-hand side of the equation is $u_I$, which represents the corresponding continuous part of the finite-element displacement solution. The second term is the nodal extension degree of the freedom vector $a_I$. $H(x)$ is the discontinuous jump function along the crack surface. The third term is the nodal extension degree of the freedom vector $b_I^{\alpha}$, and $F_{\alpha}(x)$ is the progressive stress function at the crack tip. The first term on the right can be used for all of the nodes in the model, the second term on the right is only valid for the element nodes whose shape function is cut by the crack, and the third term on the right is only valid for the element nodes whose shape function is cut at the crack tip.

ABAQUS uses the Newton–Raphson algorithm to solve nonlinear problems. By iterating the incremental step of each analysis several times, the convergence of each incremental step can be achieved, and the convergent solution of the analysis step can be obtained. In the process of iteration, ABAQUS will automatically expand or reduce the incremental step according to the convergence. The maximum circumferential tensile stress fracture criterion is built in ABAQUS, which realizes the expansion of cracks.

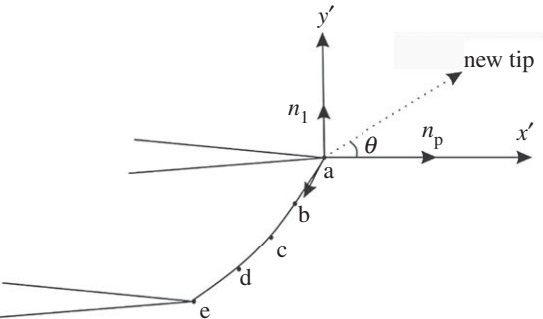

**Figure 1.** Schematic diagram of the crack.

# 3. Methods: comparative modelling of crack propagation

## 3.1. Overview

A three-dimensional elastic–plastic LRPIM for calculating fracture propagation is established. The key techniques used in this section are described: (i) based on the theory of fracture mechanics, the basic method of three-dimensional fracture propagation is established; (ii) based on the basic theory of an elastic–plastic model, the stress–strain constitutive relationship and the control equation derivation process are established; and (iii) combining fracture mechanics theory, elastic–plastic theory and the meshless LRPIM, a flow chart for calculating crack propagation in elastic–plastic materials is proposed.

## 3.2. Fracture propagation theory

### 3.2.1. Energy release rate criterion

The strain energy is stored at the fracture tip. When the fracture expands to a smaller length, the strain energy is released and the fracture expands along the direction of the maximum energy release rate [32]:

$$\frac{\partial G_\theta}{\partial \theta} = 0, \quad \frac{\partial^2 G_\theta}{\partial^2 \theta} < 0. \tag{3.1}$$

When the energy release rate reaches the critical value in some direction of the crack tip, the crack propagates:

$$(G_\theta)_{\max} = (G_\theta)_{\theta=\theta_0} = G_{IC}. \tag{3.2}$$

Through the maximum energy release rate theory, we can determine the direction of expansion and whether expansion occurs. Only the material-related parameters and critical energy release rate are needed.

For I–II tension-shear cracks, the energy release rate formula at the crack tip is as follows:

$$G = \frac{1-\nu^2}{E} K_I^2 + \frac{1-\nu^2}{E} K_{II}^2, \tag{3.3}$$

where $K_I$ and $K_{II}$ are the stress intensity factors of type I and type II cracks, respectively.

For mode I–II composite cracks, the direction of maximum energy release rate is the direction of the maximum circumferential stress. According to the theory of the maximum circumferential stress, the direction of propagation is as follows:

$$\tan\frac{\theta_0}{2} = \frac{1}{4}\left[\frac{K_I}{K_{II}} \pm \sqrt{\left(\frac{K_I}{K_{II}}\right)^2 + 8}\right], \tag{3.4}$$

where $\theta_0$ is the extension angle, or the angle between the extension direction and the $x$-axis, as shown in figure 1. In the formula, when greater than 0, take the '+' number, and when less than 0, take the '−' number.

For mode I–II composite cracks, the critical conditions under the plane strain state can be expressed by the stress intensity factors as follows:

$$K_{IC} = \cos\frac{\theta_0}{2}\left[K_I \cos^2\frac{\theta_0}{2} - \frac{3}{2}K_{II}\sin\theta_0\right]. \tag{3.5}$$

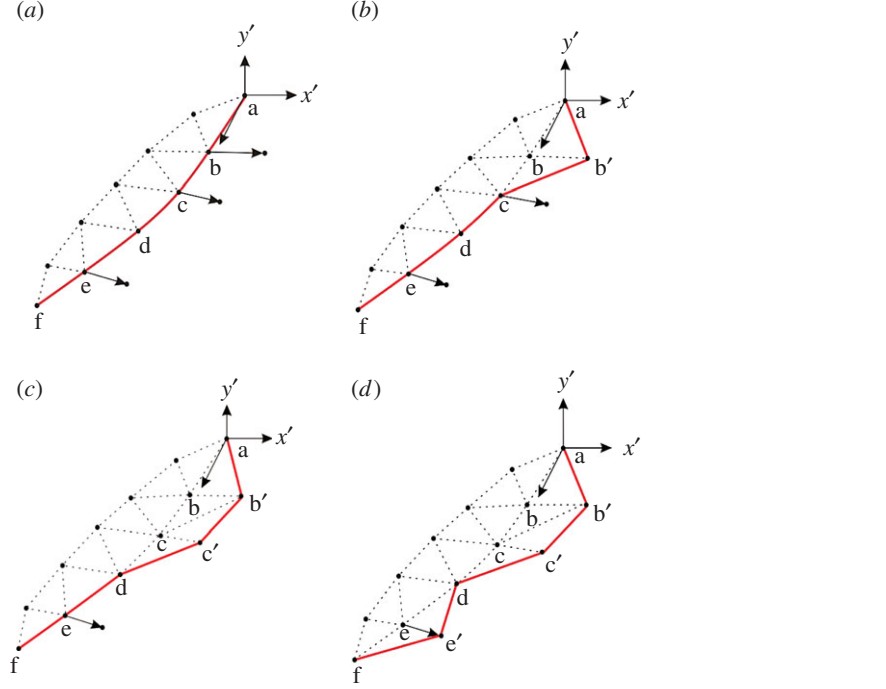

**Figure 2.** Schematic diagram of crack expansion [9]. (*a*) Judge the expandable point at the crack tip. (*b*) Realise the crack expansion of the first point. (*c*) Expand other expandable points in turn. (*d*) One step expansion completed.

### 3.2.2. Calculation of the stress intensity factor

A common method for solving the stress intensity factors is the tip displacement method. In the expression of crack tip displacement, the displacement of the upper and lower crack surfaces can be obtained by calculating the model with $\theta = \pm\pi$. The formula for the three-dimensional discontinuous displacement method for calculating the stress intensity factor at the crack front is as follows:

$$
\left.
\begin{aligned}
K_{\mathrm{I}} &= \frac{E}{8(1-\nu^2)}\sqrt{\frac{2\pi}{r}}\lim_{x\to 0}\left[u_{c2}(\theta=+\pi)-u_{c2}(\theta=-\pi)\right], \\[1mm]
K_{\mathrm{II}} &= \frac{E}{8(1-\nu^2)}\sqrt{\frac{2\pi}{r}}\lim_{x\to 0}\left[u_{c1}(\theta=+\pi)-u_{c1}(\theta=-\pi)\right] \\[1mm]
K_{\mathrm{III}} &= \frac{E}{8(1-\nu^2)}\sqrt{\frac{2\pi}{r}}\lim_{x\to 0}\left[u_{c3}(\theta=+\pi)-u_{c3}(\theta=-\pi)\right].
\end{aligned}
\right\}
\tag{3.6}
$$

and

$u_{c1}$, $u_{c2}$, $u_{c3}$ are the displacements of points in the local coordinate system and $(\theta=+\pi)$, $(\theta=-\pi)$ indicates whether the point is located on or below the fracture surface.

### 3.2.3. Crack propagation

After determining the cracking criterion, we can judge whether the tip is cracked and the direction of cracking. The next step is to determine the geometric expansion form of the crack surface.

Three-dimensional crack propagation is simulated by propagating the crack front points in the normal plane of their respective crack front. The fracture surface is simulated by a group of small triangles, and a group of lines is used to simulate the crack front.

As shown in figure 2, to propagate the fracture surface, the displacement field is calculated first, then the stress intensity factor and energy release rate of each crack front point are obtained according to the above method, including points a, b, c, d, e, f. Once a crack front point satisfies the crack propagation condition, such as b, c, e, the new positions of each crack front point are determined, like b', c', e'. Then, the crack front point information is updated, a new triangle between the new and old crack fronts is added and the crack surface information is updated. The next calculation step is carried out until no crack front point satisfies the propagation condition.

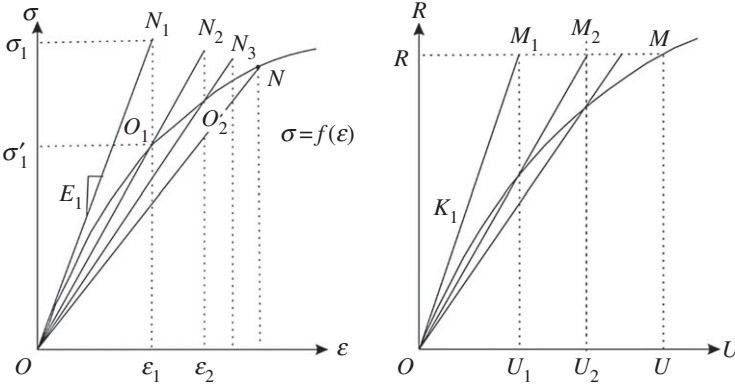

**Figure 3.** Iterative solution diagram.

As the fracture surface expands, the fracture will reach the model boundary. When the fracture surface extends to the model boundary, the position of the new tip will be corrected and set as the intersection of the extended direction and the model boundary. If the crack extends to intersect with the boundary surface, the cracks can only propagate in the boundary surface in the next step.

## 3.3. Elastoplastic calculation

In the elastoplastic calculation, the plasticity model that was used was assumed to be a elastic-perfect plastic model. The direct iteration method is used to solve the global discrete system equation. In the process of iteration, the stiffness is constantly revised for trial calculation to solve the nonlinear problem (figure 3):

$$KU = R. \tag{3.7}$$

The calculation process is as follows:

(i) in the first trial calculation, the stiffness matrix $K$ is obtained by taking the initial elastic constants $E_1$ and $v_1$;

(ii) the first approximate displacement $U_1$ is obtained under load $R$. Thus, $\varepsilon_1$ and stress $\sigma_1$ are obtained, as shown in figure 3;

(iii) because the setting of $E_1$ and $v_1$ is inaccurate, $M_1$ and $N_1$ are not on the curve. Therefore, according to $\varepsilon_1$, the corresponding stress $\sigma'_1$ is found from the functional relation $\sigma = f(\varepsilon)$ in the graph, as shown in point $O_1$:

$$\varepsilon_{(6\times1)} = B_{(6\times3n)}u_{(3n\times1)}, \tag{3.8}$$

$$B_{(6\times3n)} = \begin{bmatrix} \frac{\partial\phi_1}{\partial x} & 0 & 0 & \cdots & \frac{\partial\phi_n}{\partial x} & 0 & 0 \\ 0 & \frac{\partial\phi_1}{\partial y} & 0 & \cdots & 0 & \frac{\partial\phi_n}{\partial y} & 0 \\ 0 & 0 & \frac{\partial\phi_1}{\partial z} & \cdots & 0 & 0 & \frac{\partial\phi_n}{\partial z} \\ 0 & \frac{\partial\phi_1}{\partial z} & \frac{\partial\phi_1}{\partial y} & \cdots & 0 & \frac{\partial\phi_n}{\partial z} & \frac{\partial\phi_n}{\partial y} \\ \frac{\partial\phi_1}{\partial z} & 0 & \frac{\partial\phi_1}{\partial x} & \cdots & \frac{\partial\phi_n}{\partial z} & 0 & \frac{\partial\phi_n}{\partial x} \\ \frac{\partial\phi_1}{\partial y} & \frac{\partial\phi_1}{\partial x} & 0 & \cdots & \frac{\partial\phi_n}{\partial y} & \frac{\partial\phi_n}{\partial x} & 0 \end{bmatrix}, \tag{3.9}$$

$\phi$ is shape function constructed by LRPIM method;

(iv) the slope of secant $O\bar{O}_1$ is the secant modulus $E_2$, then, $K_2$ is obtained; and

(v) repeat steps (ii)–(iv) to obtain $M_2$, $N_2$, $O_2$, $M_3$, $N_3$, $O_3$ and so on. When the displacement value of the last two iterations is close and the error is less than the allowable value, the calculation is complete:

$$K_1U_1 = R, \tag{3.10}$$

$$K(U_{i-1})U_i = R \tag{3.11}$$

and

$$\left\|\frac{a^{n+1} - a^n}{a^{n+1}}\right\| < \varepsilon. \tag{3.12}$$

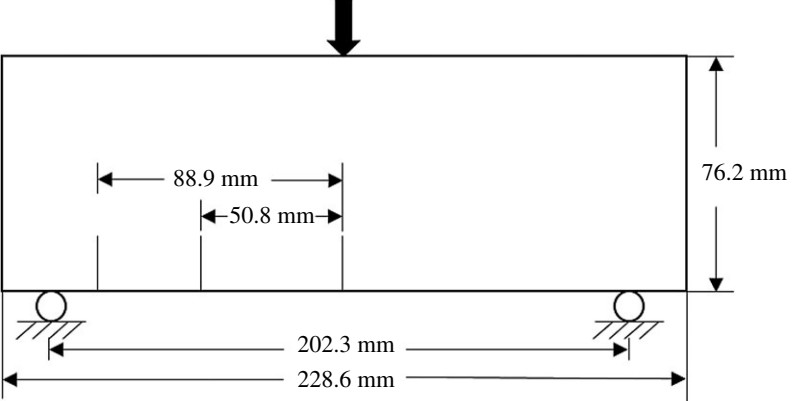

**Figure 4.** Model schematic.

## 3.4. Crack propagation calculation for elastic–plastic materials

The emphasis of this paper is to combine elastic–plastic calculation, crack propagation and meshless methods to solve model expansion. The crack propagation process is as follows:

(i) set the initial conditions, including the arrangement of nodes, defining material parameters, imposing boundary conditions, imposing loads and so on;

(ii) enter the iteration step calculation and converge through several iteration steps. The stress–strain state of the whole model is obtained by solving the problem; and

(iii) the stress intensity factor is calculated by the stress state at the crack tip. According to the fracture criterion, the extension direction is enough and extended according to the corresponding geometric expansion method until the model does not extend.

# 4. Results: case studies

## 4.1. Mode test: fracture in concrete

In this section, single-edge notched, three-point-bend specimens made from concrete are simulated (figure 4). The material properties measured in the experiments were tensile strength, $\sigma_b = 13.1$ MPa; elastic modulus, $E = 34.48$ GPa; and Poisson's ratio, $v = 0.2$. The density was assumed to be 2400 kg m$^{-3}$, cohesion to be 33 MPa, and internal friction angle to be 40°.

The experiments involved a mixed-mode crack propagation in a three-point-bend specimen with an initial notch offset from the midspan. The specimen geometry is shown in figure 4. The location of the notch is described in terms of a normalized parameter $\gamma$, where $\gamma$ is the distance from the midspan to the notch divided by the distance from the midspan to a support. The following parameters were used in the experiments: length = 228.6 mm; height = 76.2 mm; thickness = 25.4 mm; distance between supports = 203.2 mm; notch length = 19.0 mm; and notch location of $\gamma = 0$, $\gamma = 0.5$ to $\gamma = 0.875$.

The experimental results were reported by John and Shah [33]. The calculated results of the three models are shown in figure 5. The angles between the straight line and the vertical direction fitted by the cracking direction are 0°, 22° and 30°.

## 4.2. Radial basis point interpolation method simulation results

According to the literature, three groups of numerical experiments were designed and the expansion step was set to 2 mm. When the crack tip satisfies the cracking criterion, it expands. The experimental results are as follows.

### 4.2.1. Mode I fracture

When the fracture surface is located in the middle of the bottom of the three-point bending beam, it is a type I fracture. The initial vertical displacement in the middle of the upper surface is $-1 \times 10^{-3}$ mm, and the x-direction displacement and stress nephogram of the model are obtained (figures 6 and 7).

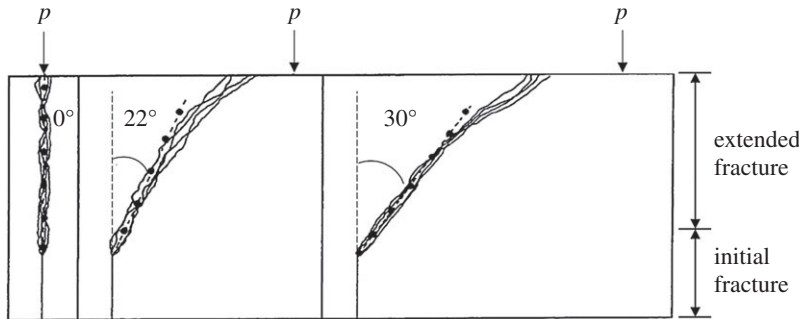

**Figure 5.** Physical model test results.

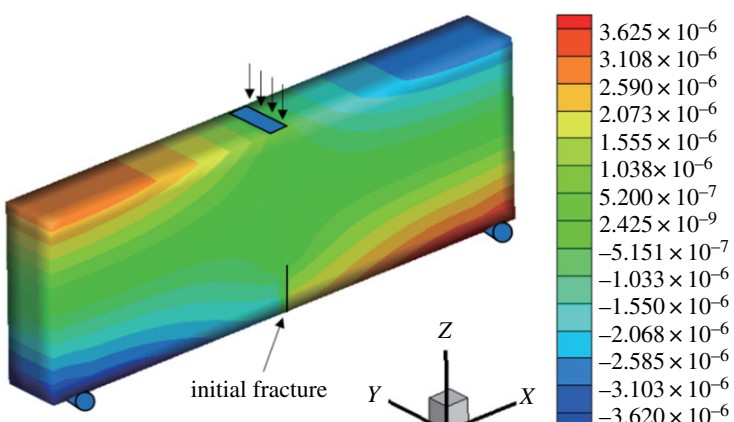

**Figure 6.** Horizontal displacement nephogram ($\gamma = 0$) (m).

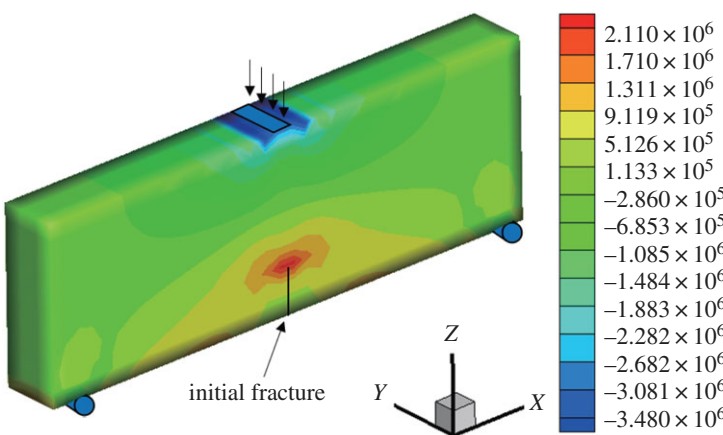

**Figure 7.** Horizontal stress nephogram ($\gamma = 0$) (Pa).

The displacement is obviously discontinuous, and the stress nephogram in the $x$ direction shows that there is stress concentration at the tip, resulting in a larger tensile stress. The horizontal displacement value is $-3.625 \times 10^{-3}$ to $3.62 \times 10^{-3}$ mm, and the horizontal stress value is $-3.48$ to $2.11$ MPa.

When the stress–strain state of the crack tip satisfies the fracture criterion, the crack surface expands. After several steps, the fracture tip basically extends along the vertical direction, and the angle between the vertical direction and the fracture surface extension direction is $0°$, which is in line with the expectation (figure 8).

### 4.2.2. Mix mode fracture

When $\gamma = 0.5$, the initial vertical displacement value applied in the middle of the upper surface is $-1 \times 10^{-2}$ mm, and the horizontal displacement and stress nephograms of the model are shown

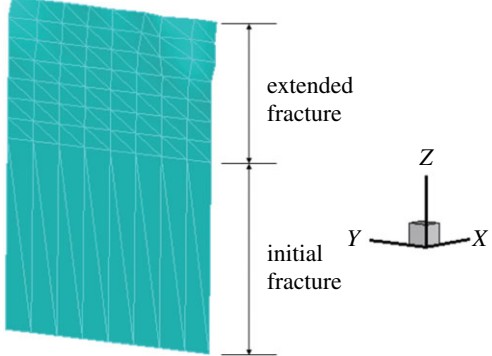

**Figure 8.** Fracture expansion diagram ($\gamma = 0$).

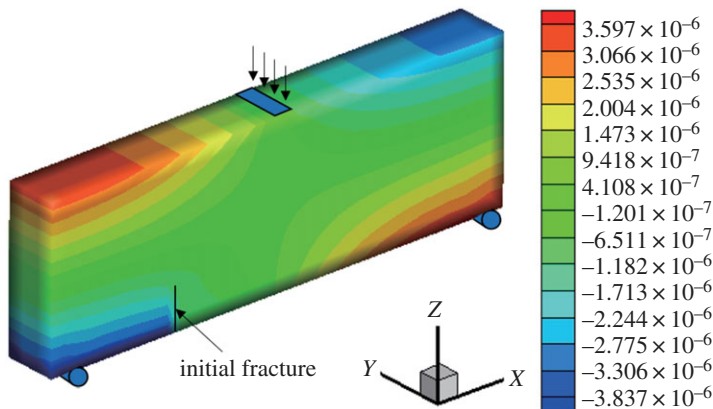

**Figure 9.** Horizontal displacement nephogram ($\gamma = 0.5$) (m).

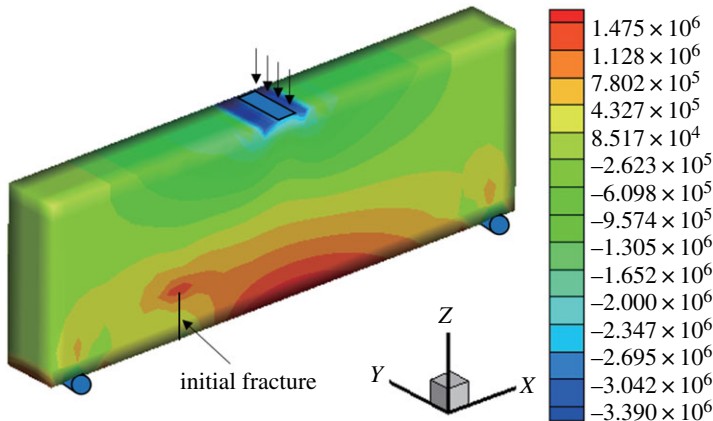

**Figure 10.** Horizontal stress nephogram ($\gamma = 0.5$) (Pa).

(figures 9 and 10). The horizontal normal displacement occurs discontinuously at the position of the fracture surface, and the stress at the tip of the fracture surface is concentrated. It can be observed that the displacement of the fissure surface is obviously discontinuous, and the stress cloud diagram shows that there is stress concentration at the tip, resulting in a larger tensile stress. The maximum horizontal displacement is $-3.837 \times 10^{-3}$ to $3.597 \times 10^{-3}$ mm, and the horizontal stress is $-3.39$ to $1.475$ MPa.

In the process of fracture surface expansion, the expansion path is not smooth, but the final expansion direction is close to the experimental results (figures 7, 8 and 11). The angle between the vertical direction and the fracture surface is 25°. It can be noted that the general direction of the extension is correct, but the LRPIM method is unstable for solving the stress intensity factor at the tip, which greatly deviates from the exact value.

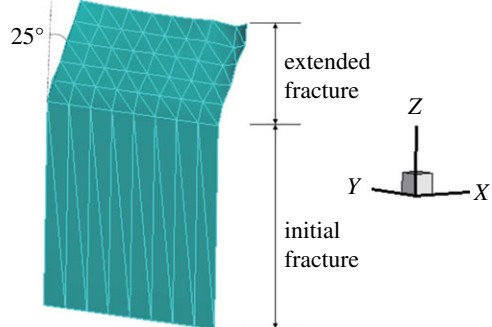

**Figure 11.** Fracture expansion diagram ($\gamma = 0.5$).

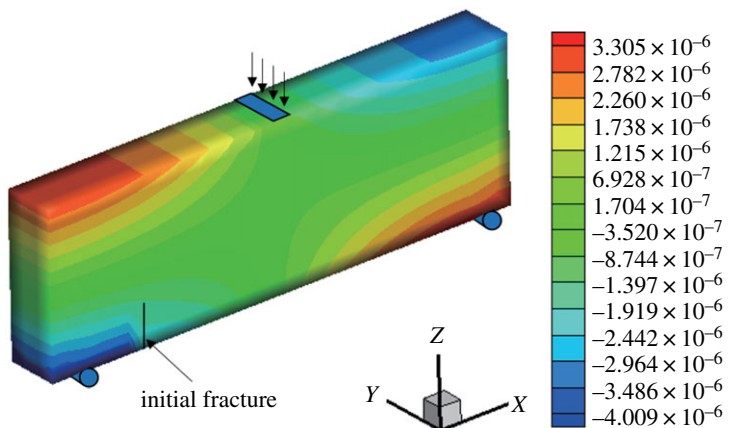

**Figure 12.** Horizontal displacement nephogram ($\gamma = 0.7$) (m).

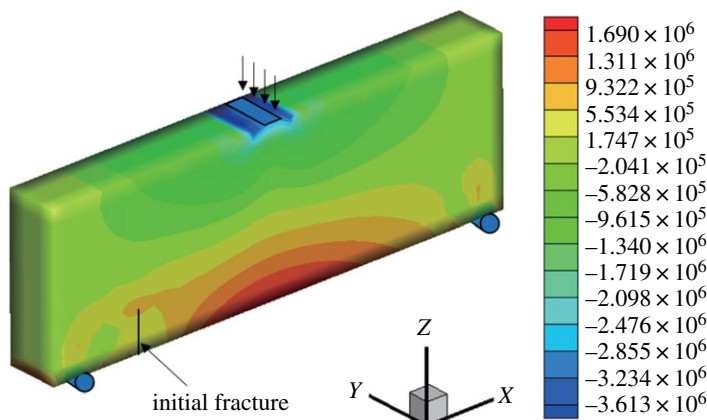

**Figure 13.** Horizontal stress nephogram ($\gamma = 0.7$) (Pa).

When $\gamma = 0.7$, the initial vertical displacement value in the middle of the upper surface is $-1 \times 10^{-2}$ mm and the horizontal displacement and stress nephograms of the model are shown in figures 12 and 13. It can be seen that the displacement is obviously discontinuous, and the stress nephogram in the horizontal direction shows that there is a stress concentration at the tip, resulting in a larger tensile stress. The maximum horizontal displacement is $-4.009 \times 10^{-3}$ to $3.305 \times 10^{-3}$ mm, and the horizontal stress is $-3.613$ to $1.69$ MPa.

Like the above two models, the extension path is not smooth (figures 7, 8 and 14); however, the final extension direction is close to the experimental results, and the expansion angle of the fracture surface is 28°.

## 4.3. eXtended finite-element simulation results

XFEM is used to simulate the crack growth of a three-point bending beam and the maximum circumferential tensile stress fracture criterion is employed.

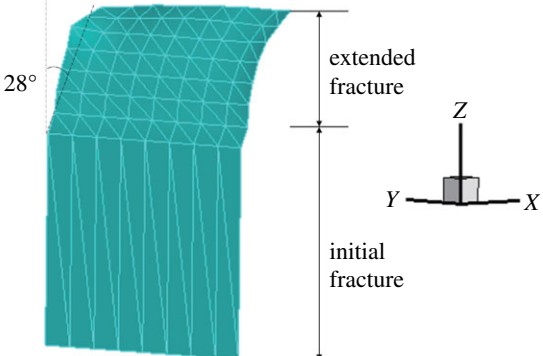

**Figure 14.** Fracture expansion diagram ($\gamma = 0.7$).

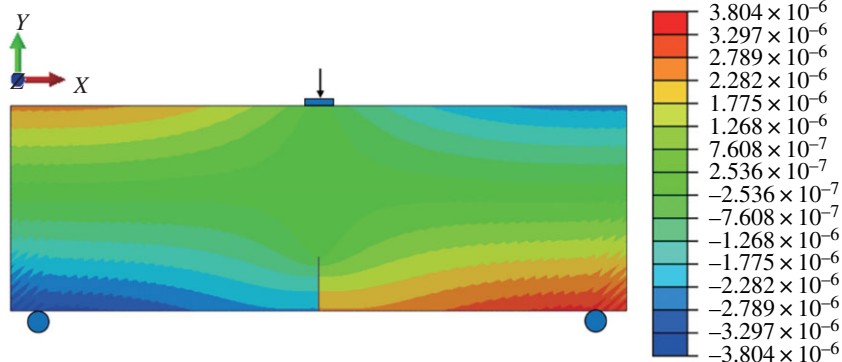

**Figure 15.** Horizontal displacement nephogram ($\gamma = 0$) (m).

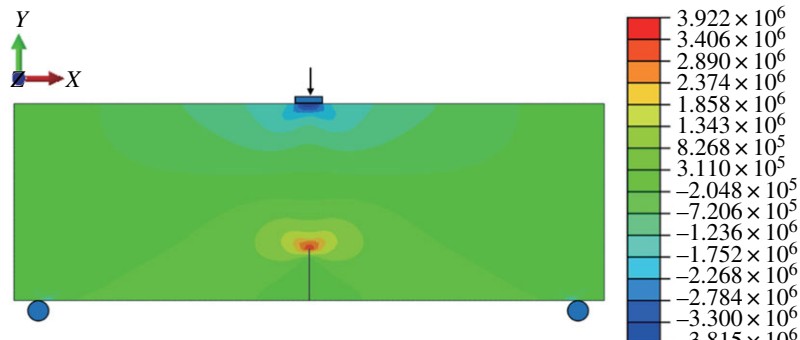

**Figure 16.** Horizontal stress nephogram ($\gamma = 0$) (Pa).

### 4.3.1. Mode I fracture

When the fracture is in the middle of the model bottom, the stress–strain nephogram before expansion is shown in figures 15 and 16. The displacement at the fracture surface is discontinuous and stress concentration occurs at the top of the fracture surface. The path of fracture propagation is vertical upwards (figure 17), and the angle between the direction of fracture propagation and vertical is 0°. The horizontal displacement value is $-3.804 \times 10^{-3}$ to $3.804 \times 10^{-3}$ mm, and the horizontal stress value is $-3.815$ to $3.922$ MPa.

### 4.3.2. Mix mode fracture

When $\gamma = 0.5$, the stress–strain nephogram is shown as below (figures 18 and 19). A displacement discontinuity and stress concentration occur and the angle between the fracture surface and the

<c13

<croyalsocietypublishing.org/journal/rsos    R. Soc. open sci. 6: 190543

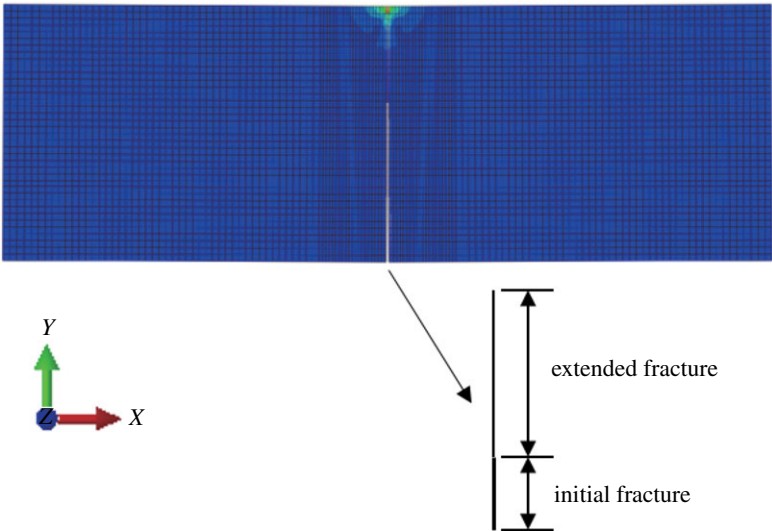

**Figure 17.** Fracture expansion diagram ($\gamma = 0$).

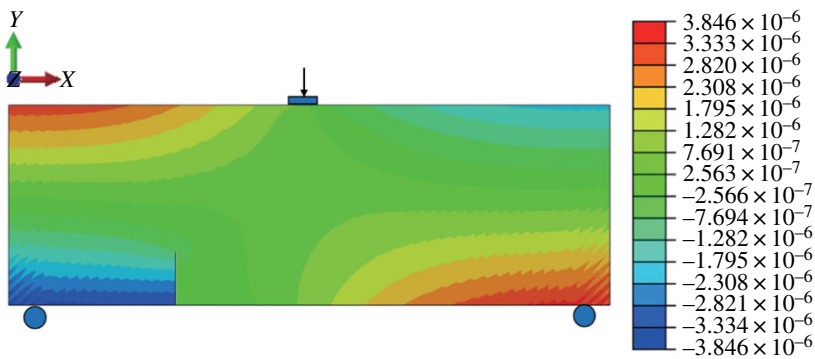

**Figure 18.** Horizontal displacement nephogram ($\gamma = 0.5$) (m).

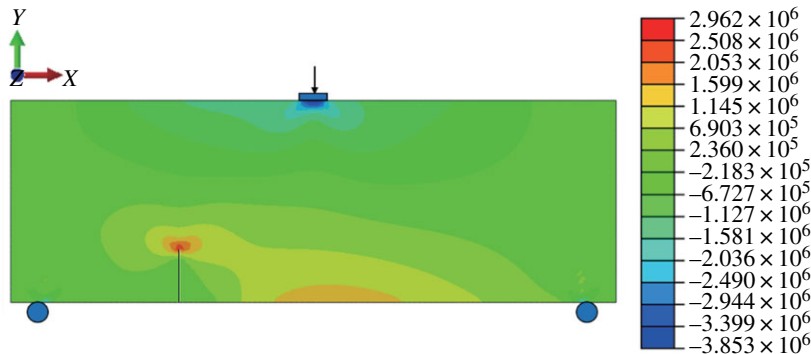

**Figure 19.** Horizontal stress nephogram ($\gamma = 0.5$) (Pa).

vertical direction is 21.1° after expansion (figure 20). The experimental results are relatively small compared with the physical model tests, but close. The horizontal displacement value is $-3.846 \times 10^{-3}$ to $3.846 \times 10^{-3}$ mm, and the horizontal stress value is $-3.853$ to $2.962$ MPa.

When $\gamma = 0.7$, the stress-displacement nephogram before the fracture surface does not expand, as shown below, and a displacement discontinuity and stress concentration occur (figures 21 and 22). The angle between the expanded fracture surface and the vertical direction is 27.4° (figure 23), which is very close to the results of physical experiments. The horizontal displacement value is $-3.666 \times 10^{-3}$ to $3.415 \times 10^{-3}$ mm, and the maximum horizontal stress value is $-3.975$ to $3.157$ MPa.

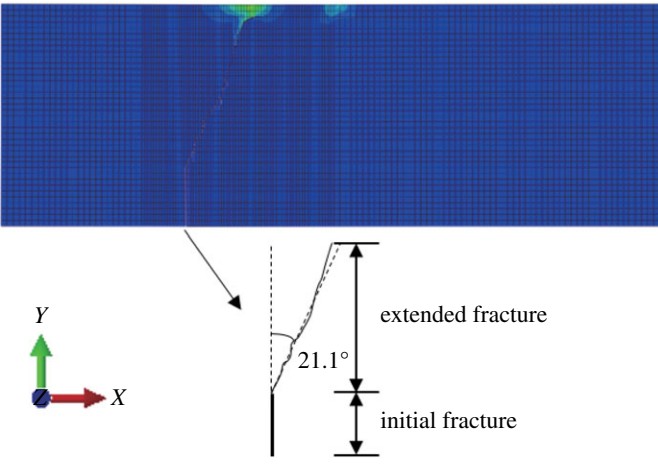

**Figure 20.** Fracture expansion diagram ($\gamma = 0.5$).

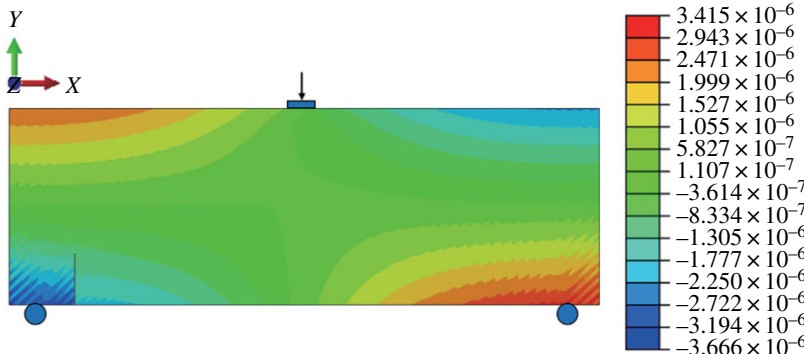

**Figure 21.** Horizontal displacement nephogram ($\gamma = 0.7$) (m).

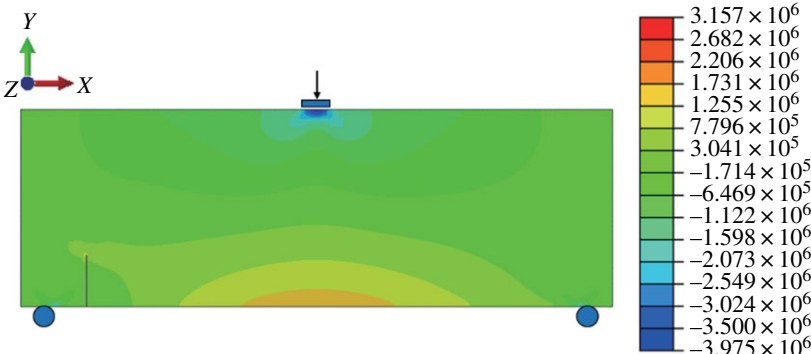

**Figure 22.** Horizontal stress nephogram ($\gamma = 0.7$) (Pa).

## 5. Discussion

In this section, we analyse (i) physical experiments versus LRPIM when using an elastic–plastic model and (ii) LRPIM versus XFEM (using the physical experiments as the baseline).

### 5.1. Physical experiments versus local radial basis point interpolation method when using an elastic–plastic model

Using LRPIM, the crack propagation patterns of three-dimensional concrete structures under elastic–plastic conditions are obtained. The increase of the angle between the fissure and the vertical direction

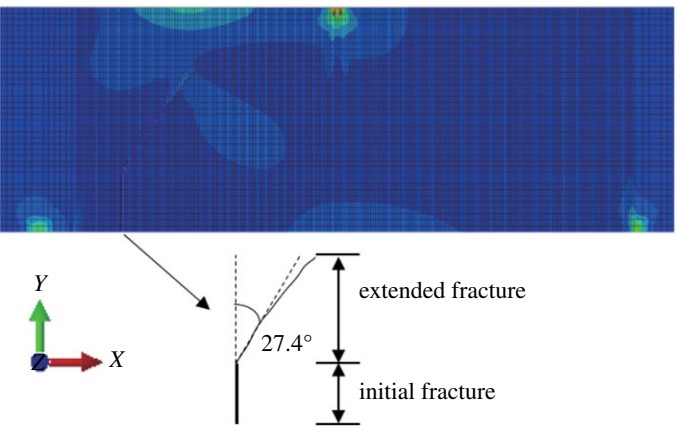

**Figure 23.** Fracture expansion diagram ($\gamma = 0.7$).

**Table 1.** The angle between fracture surface and vertical surface.

| | the angle between fracture surface and vertical surface (°) | | |
|---|---|---|---|
| method | $\gamma = 0$ | $\gamma = 0.5$ | $\gamma = 0.7$ |
| ABAQUS | 0 | 21.1 | 27.4 |
| GeoMFree | 0 | 25 | 28 |
| physical experiment | 0 | 22 | 30 |

is the same as the physical model test (table 1). It is undeniable that there are some errors between the numerical value and the physical test, which may be related to the integration domain, support domain and other factors. Through this method, we can also obtain stress, strain information during the test.

In the fracture expansion simulation, the fracture front edge is discretized into a series of points, then, the direction of the fracture expansion is calculated in a plane with a normal relationship to the front edge of the fracture, and whether the fracture expands or not is judged. Although this method can reflect the characteristics of fracture propagation in a three-dimensional model, it is still limited to the content of two-dimensional fracture mechanics and requires further exploration in theory.

The shorter the theoretical expansion step, the better the morphological characteristics of the fracture surface can be reflected. However, the computational cost will significantly increase. Meshless methods are low efficiency numerical methods. Therefore, only a certain small value can be used for the extended step size, which also has a certain impact on the accuracy of the calculation results.

In this paper, the elastic–plastic constitutive model and the three-dimensional LRPIM are combined to calculate the failure process of elastic–plastic materials. As a continuously improved numerical method, the meshless method still demands refining. The integral domain, support domain, weight function and even layout of field nodes will affect the results. The nonlinear correlation makes it difficult to obtain sufficiently accurate results using the LRPIM. For models that need to obtain high-precision results, further research is needed.

## 5.2. Radial basis point interpolation method versus eXtended finite-element method (using the physical experiments as the baseline)

XFEM is a mature numerical method and it has been embedded in commercial software. In this paper, the crack propagation patterns of concrete structures under two-dimensional elastic–plastic conditions are obtained using this method. With increasing $\gamma$, the angle between the fissure and the vertical direction increases, the same as for the physical model test and the numerical value of the angle is very close to that of the physical model test.

Similar to the LRPIM method, the stress-displacement information in the process of fracture propagation can also be obtained (tables 2–4), which verifies the correctness of the method. XFEM is

**Table 2.** Comparative table of calculation results ($\gamma = 0$).

| method | horizontal displacement (mm) | | maximum horizontal tensile stress (MPa) |
| --- | --- | --- | --- |
| | minimum value | maximum value | |
| ABAQUS | −3.804 | 3.804 | 3.922 |
| GeoMFree | −4.138 | 4.143 | 3.88 |
| error | 0.087 | 0.089 | 0.011 |

**Table 3.** Comparative table of calculation results ($\gamma = 0.5$).

| method | horizontal displacement (mm) | | maximum horizontal tensile stress (MPa) |
| --- | --- | --- | --- |
| | minimum value | maximum value | |
| ABAQUS | −3.846 | 3.846 | 2.962 |
| GeoMFree | −4.333 | 4.092 | 1.962 |
| error | 0.126 | 0.064 | 0.337 |

**Table 4.** Comparative table of calculation results ($\gamma = 0.7$).

| method | horizontal displacement (mm) | | maximum horizontal tensile stress (MPa) |
| --- | --- | --- | --- |
| | minimum value | maximum value | |
| ABAQUS | −3.666 | 3.415 | 3.157 |
| GeoMFree | −4.496 | 3.792 | 2.068 |
| error | 0.226 | 0.11 | 0.345 |

a grid-based numerical method, so it cannot arbitrarily refine the computational domain like the meshless method to obtain greater accuracy. The main work in this paper is numerical analysis, no validation is provided.

# 6. Conclusion

In this paper, the elastic–plastic constitutive model is combined with the three-dimensional LRPIM and is employed in a comparative investigation of crack propagation. Numerical modelling for concrete structures using the meshfree LRPIM and XFEM is compared. The results of the two methods are consistent with that of physical experiments. As a mature numerical method, XFEM obtains closer calculation results than LRPIM. However, because XFEM is grid-based, it inevitably faces problems brought by the grid, especially for model calculation under elastic–plastic conditions. LRPIM does not have to consider these problems, because it is meshless. In addition to that, LRPIM has fast convergence speed, good smoothness, no post-processing and no volume locking. It has incomparable advantages over the finite element and finite difference methods, and it is a very promising numerical method.

Data accessibility. Our data have been deposited at the Dryad Digital Repository: https://doi.org/10.5061/dryad.kprr4xh0n [34].

Authors' contributions. Y.L. and N.X. designed the study and wrote the manuscript. Y.L. calculated the model. G.M. gave guidance on the writing and translation of the paper and J.T. used XFEM to calculate the comparison model. All the authors gave their final approval for publication.

Competing interests. We declare we have no competing interests.

Funding. This work was supported by the Natural Science Foundation of China (grant nos. 11602235 and 41772326), and the Fundamental Research Funds for the Central Universities (grant nos. 2652018091, 2652018107 and 2652018109).

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
