## [Reviewer comments · Royal Society Open Science]

Review History

RSOS-190543.R0 (Original submission)

Review form: Reviewer 1

Is the manuscript scientifically sound in its present form?

No

Are the interpretations and conclusions justified by the results?

Yes

Is the language acceptable?

Yes

Is it clear how to access all supporting data?

Not Applicable

Do you have any ethical concerns with this paper?

No

Have you any concerns about statistical analyses in this paper?

No

Recommendation?

Major revision is needed (please make suggestions in comments)

Comments to the Author(s)

RSOS-190543 review comments

This paper presents interesting numerical approaches of modeling crack propagation in brittle materials. The combined usage of local radial basis point Interpolation method and linear fracture mechanics theory for modeling crack propagating processes is innovative. However, some theoretical descriptions were not well presented, and lots of references were missing. There were also some minor issues in typos and formatting. I thus suggest a major revision to improve the quality and readability of this manuscript. Detailed comments are as the following:

1. Page 4, Line 73

It is mentioned that "Unlike the global weak form, this method does not require the background.."

To my knowledge, the weak forms for meshfree, FEM, and XFEM are the same, and the differences are at the spatial discretization and interpolation approximation. It is recommended to double check whether you need a different weak form for LRPIM.

In addition, related references should be added since LRPIM has been quite popular.

2. Page 6, Line 108. Failure criterion used for XFEM was not indicated

It is also recommended to present the failure criterion used in XFEM modeling.

3. Page 11, Line 193. A detailed description of Fig. 2 is needed

Since Fig. 2 has sub-plots of A,B,C,D, more detailed descriptions on those figures are needed.

4. Page 12, Line 211. The description on elastoplastic theory is incomplete and easily gets misunderstandings

Elastoplastic is generally used to quantify the stress-strain relations of materials with clear hardening or softening behaviors. Thus, a phenomenon-based yielding criterion and flow rule are required to generate an elasto-plastic matrix. The authors presents a Newton-Raphson approach of solving non-linear equations instead of a elastoplastic theory. The title of 3.3 or the content should be rewritten to have a consistent description.

By the side, it should be well documented that how to derive strains. Is there plastic strains? If only a Griffith type of failure criterion was applied, it seems that there is only elastic strains existing in non-fracture domain.

5. Some minor issues

1) Units for nephogram plots should be added to the captions. (e.g., Fig. 6)

2) Page 2, Line 8

"omparative" change to "Comparative"

3) Page 8, Line 136

4) "vectoral" change to "vector aI"

5) Page 9, Line 149

“In this section, we briefly introduce the basic principles of the LRPIM and XFEM.” should be deleted or replaced by a relevant sentence.

6) Page 9, Line 161

References for fracture propagation theory should be added.

7) Page 10, Line 180

There are some symbols which do not appear correctly.

8) Page 11, Line 189

uc1, uc1, and uc3 should defined.

9) Page 14, Line 231

The sentence is not complete.

10) Page 21, Line 321

“mpa” change to “MPa”

11) Page 25, Line 360

Table2-Table 4 only give information of stress-displacement information instead of stress-strain information.

12) The format of references should be adjusted for this journal.

END

Review form: Reviewer 2

Is the manuscript scientifically sound in its present form?

Yes

Are the interpretations and conclusions justified by the results?

Yes

Is the language acceptable?

No

Is it clear how to access all supporting data?

Yes

Do you have any ethical concerns with this paper?

No

Have you any concerns about statistical analyses in this paper?

No

Recommendation?

Major revision is needed (please make suggestions in comments)

Comments to the Author(s)

This paper presents so much theoretical literature. The actual work is very limited and confined to the numerical analysis only. No validation is provided. Please mention this limitation in the discussion or conclusion section. Most of the references are more than 10 years old. In fact, more than 5-year old references is very often discouraged. Delete some old ref and include some new. One example, Modeling Crack Propagation in Asphalt Concrete Overlay using Extended Finite Element Model, ASCE Journal of Materials in Civil Engineering, 29(5). The flow of the paper is good, but, please write in passive format. Lines 149, 243: 'We'???

Review form: Reviewer 3 (Sebnem Ozupek)

Is the manuscript scientifically sound in its present form?

Yes

Are the interpretations and conclusions justified by the results?

Yes

Is the language acceptable?

Yes

Is it clear how to access all supporting data?

Yes

Do you have any ethical concerns with this paper?

No

Have you any concerns about statistical analyses in this paper?

I do not feel qualified to assess the statistics

Recommendation?

Accept with minor revision (please list in comments)

Comments to the Author(s)

1. line 175: "maximum circumferential normal stress" needs to be corrected as "maximum circumferential stress" or "maximum normal stress".
2. line 181:incomplete sentence
3. Equation (17): K_i should be K_I
4. Section 3.3 talks about elastoplastic theory: what is the plasticity model used in the paper? There are several models and it is not clear which one was assumed to hold in the calculations.
5. line 271: K_1 should be E_1 ?
6. lines 215 through 229 need to be reorganized. There are several badly written phrases that make the sequence 1 through 6 not understandable.
7. line 231: incomplete sentence

8. line 245: what is "f"?
9. line 246: the analysis is static, why density is needed?
10. line 250: Should Figure 12 be Figure 4?
11. line 266 : explain why $-1e-3$ mm was selected as the displacement?
12. line 267: incomplete sentence
13. lines 273-274: explain or rephrase "after several expansions, the fracture surface basically expands...".

Decision letter (RSOS-190543.R0)

10-Jul-2019

Dear Dr Li,

The editors assigned to your paper ("Comparative Modelling of Crack Propagation in Elastic-Plastic Materials Using Meshfree LRPIM and XFEM") have now received comments from reviewers. We would like you to revise your paper in accordance with the referee and Associate Editor suggestions which can be found below (not including confidential reports to the Editor). Please note this decision does not guarantee eventual acceptance.

Please submit a copy of your revised paper before 02-Aug-2019. Please note that the revision deadline will expire at 00.00am on this date. If we do not hear from you within this time then it will be assumed that the paper has been withdrawn. In exceptional circumstances, extensions may be possible if agreed with the Editorial Office in advance. We do not allow multiple rounds of revision so we urge you to make every effort to fully address all of the comments at this stage. If deemed necessary by the Editors, your manuscript will be sent back to one or more of the original reviewers for assessment. If the original reviewers are not available, we may invite new reviewers.

- Data accessibility

<http://datadryad.org/submit?journalID=RSOS&manu=RSOS-190543>

- Competing interests

- Authors' contributions

- Acknowledgements

- Funding statement

on behalf of R. Kerry Rowe (Subject Editor)
 openscience@royalsociety.org

Comments to Author:

Reviewers' Comments to Author:

Reviewer: 1

Comments to the Author(s)

RSOS-190543 review comments

This paper presents interesting numerical approaches of modeling crack propagation in brittle materials. The combined usage of local radial basis point Interpolation method and linear fracture mechanics theory for modeling crack propagating processes is innovative. However, some theoretical descriptions were not well presented, and lots of references were missing. There were also some minor issues in typos and formatting. I thus suggest a major revision to improve the quality and readability of this manuscript. Detailed comments are as the following:

1. Page 4, Line 73

It is mentioned that "Unlike the global weak form, this method does not require the background.."

To my knowledge, the weak forms for meshfree, FEM, and XFEM are the same, and the differences are at the spatial discretization and interpolation approximation. It is recommended to double check whether you need a different weak form for LRPIM.

In addition, related references should be added since LRPIM has been quite popular.

2. Page 6, Line 108. Failure criterion used for XFEM was not indicated

It is also recommended to present the failure criterion used in XFEM modeling.

3. Page 11, Line 193. A detailed description of Fig. 2 is needed

Since Fig. 2 has sub-plots of A,B,C,D, more detailed descriptions on those figures are needed.

4. Page 12, Line 211. The description on elastoplastic theory is incomplete and easily gets misunderstandings

Elastoplastic is generally used to quantify the stress-strain relations of materials with clear hardening or softening behaviors. Thus, a phenomenon-based yielding criterion and flow rule are required to generate an elasto-plastic matrix. The authors presents a Newton-Raphson approach of solving non-linear equations instead of a elastoplastic theory. The title of 3.3 or the content should be rewritten to have a consistent description.

By the side, it should be well documented that how to derive strains. Is there plastic strains? If only a Griffith type of failure criterion was applied, it seems that there is only elastic strains existing in non-fracture domain.

5. Some minor issues

- 1) Units for nephogram plots should be added to the captions. (e.g., Fig. 6)
- 2) Page 2, Line 8
“omparative” change to “Comparative”
- 3) Page 8, Line 136
- 4) “vectoral” change to “vector al”
- 5) Page 9, Line 149
“In this section, we briefly introduce the basic principles of the LRPIM and XFEM.” should be deleted or replaced by a relevant sentence.
- 6) Page 9, Line 161
References for fracture propagation theory should be added.
- 7) Page 10, Line 180
There are some symbols which do not appear correctly.
- 8) Page 11, Line 189
uc1, uc1, and uc3 should defined.
- 9) Page 14, Line 231
The sentence is not complete.
- 10) Page 21, Line 321
“mpa” change to “MPa”
- 11) Page 25, Line 360
Table2-Table 4 only give information of stress-displacement information instead of stress-strain information.
- 12) The format of references should be adjusted for this journal.

END

Reviewer: 2

Comments to the Author(s)

This paper presents so much theoretical literature. The actual work is very limited and confined to the numerical analysis only. No validation is provided. Please mention this limitation in the discussion or conclusion section. Most of the references are more than 10 years old. In fact, more than 5-year old references is very often discouraged. Delete some old ref and include some new. One example, Modeling Crack Propagation in Asphalt Concrete Overlay using Extended Finite Element Model, ASCE Journal of Materials in Civil Engineering, 29(5). The flow of the paper is good, but, please write in passive format. Lines 149, 243: ‘We’???

Reviewer: 3

Comments to the Author(s)

1. line 175: "maximum circumferential normal stress" needs to be corrected as "maximum circumferential stress" or "maximum normal stress".
2. line 181: incomplete sentence
3. Equation (17): K_i should be K_I
4. Section 3.3 talks about elastoplastic theory: what is the plasticity model used in the paper? There are several models and it is not clear which one was assumed to hold in the calculations.
5. line 271: K_1 should be E_1 ?
6. lines 215 through 229 need to be reorganized. There are several badly written phrases that make the sequence 1 through 6 not understandable.
7. line 231: incomplete sentence
8. line 245: what is "f"?
9. line 246: the analysis is static, why density is needed?
10. line 250: Should Figure 12 be Figure 4?
11. line 266 : explain why $-1e-3$ mm was selected as the displacement?
12. line 267: incomplete sentence
13. lines 273-274: explain or rephrase "after several expansions, the fracture surface basically expands...".

Author's Response to Decision Letter for (RSOS-190543.R0)

See Appendix A.

RSOS-190543.R1 (Revision)

Review form: Reviewer 1

Is the manuscript scientifically sound in its present form?

Yes

Are the interpretations and conclusions justified by the results?

Yes

Is the language acceptable?

Yes

Do you have any ethical concerns with this paper?

No

Have you any concerns about statistical analyses in this paper?

No

Recommendation?

Accept as is

Comments to the Author(s)

Thanks for making necessary modifications, now the paper deserves to be published by RSOS.

Review form: Reviewer 2

Is the manuscript scientifically sound in its present form?

Yes

Are the interpretations and conclusions justified by the results?

Yes

Is the language acceptable?

Yes

Do you have any ethical concerns with this paper?

No

Have you any concerns about statistical analyses in this paper?

No

Recommendation?

Accept as is

Comments to the Author(s)

Good work!!

Review form: Reviewer 3 (Sebnem Ozupek)

Is the manuscript scientifically sound in its present form?

Yes

Are the interpretations and conclusions justified by the results?

Yes

Is the language acceptable?

Yes

Do you have any ethical concerns with this paper?

No

Have you any concerns about statistical analyses in this paper?

No

Recommendation?

Accept with minor revision (please list in comments)

Comments to the Author(s)

Items 4) and 6) in reviewer's list of comments to the original submission were not properly addressed. In particular, comment 4) which asked the authors to clearly indicate the plasticity model that was used was not answered. The reviewer believes that the authors assumed elastic-perfectly plastic model. If this is true it needs to be stated in the text.

Decision letter (RSOS-190543.R1)

03-Sep-2019

Dear Dr Li:

On behalf of the Editors, I am pleased to inform you that your Manuscript RSOS-190543.R1 entitled "Comparative Modelling of Crack Propagation in Elastic-Plastic Materials Using Meshfree LRPIM and XFEM" has been accepted for publication in Royal Society Open Science subject to minor revision in accordance with the referee suggestions. Please find the referees' comments at the end of this email.

The reviewers and Subject Editor have recommended publication, but also suggest some minor revisions to your manuscript. Therefore, I invite you to respond to the comments and revise your manuscript.

- Ethics statement

- Data accessibility

It is a condition of publication that all supporting data are made available either as supplementary information or preferably in a suitable permanent repository. The data accessibility section should state where the article's supporting data can be accessed. This section should also include details, where possible of where to access other relevant research materials such as statistical tools, protocols, software etc can be accessed. If the data has been deposited in an external repository this section should list the database, accession number and link to the DOI for all data from the article that has been made publicly available. Data sets that have been

deposited in an external repository and have a DOI should also be appropriately cited in the manuscript and included in the reference list.

If you wish to submit your supporting data or code to Dryad (<http://datadryad.org/>), or modify your current submission to dryad, please use the following link:
<http://datadryad.org/submit?journalID=RSOS&manu=RSOS-190543.R1>

- **Competing interests**

- **Authors' contributions**

- **Acknowledgements**

- **Funding statement**

Because the schedule for publication is very tight, it is a condition of publication that you submit the revised version of your manuscript before 12-Sep-2019. Please note that the revision deadline will expire at 00.00am on this date. If you do not think you will be able to meet this date please let me know immediately.

When submitting your revised manuscript, you will be able to respond to the comments made by the referees and upload a file "Response to Referees" in "Section 6 - File Upload". You can use this

to document any changes you make to the original manuscript. In order to expedite the processing of the revised manuscript, please be as specific as possible in your response to the referees.

on behalf of Prof R. Kerry Rowe (Subject Editor)
openscience@royalsociety.org

Associate Editor Comments to Author:

Thank you for your attention to the reviewers' comments. The general view seems to be that your manuscript is nearly 'over the line' in terms of readiness for publication. However, one of the reviewers notes:

"items 4) and 6) in reviewer's list of comments to the original submission were not properly addressed. In particular, comment 4) which asked the authors to clearly indicate the plasticity model that was used was not answered. The reviewer believes that the authors assumed elastic-perfectly plastic model. If this is true it needs to be stated in the text."

Given their comments, the Editors would like you to look again at those concerns and properly

respond to them as the reviewer asks. Please ensure you include both a point-by-point response and a tracked-changes version of the paper with your revision.

Reviewer comments to Author:

Reviewer: 2

Comments to the Author(s)

Good work!!

Reviewer: 1

Comments to the Author(s)

Thanks for making necessary modifications, now the paper deserves to be published by RSOS.

Reviewer: 3

Comments to the Author(s)

Items 4) and 6) in reviewer's list of comments to the original submission were not properly addressed. In particular, comment 4) which asked the authors to clearly indicate the plasticity model that was used was not answered. The reviewer believes that the authors assumed elastic-perfectly plastic model. If this is true it needs to be stated in the text.

Author's Response to Decision Letter for (RSOS-190543.R1)

See Appendix B.

Decision letter (RSOS-190543.R2)

22-Oct-2019

Dear Dr Li,

I am pleased to inform you that your manuscript entitled "Comparative Modelling of Crack Propagation in Elastic-Plastic Materials Using Meshfree LRPIM and XFEM" is now accepted for publication in Royal Society Open Science.

Kind regards,
Lianne Parkhouse
Editorial Coordinator
Royal Society Open Science
openscience@royalsociety.org

on behalf of the Associate Editor, and Professor R. Kerry Rowe (Subject Editor)
openscience@royalsociety.org

Appendix A

Reviewer 1

Thank you very much for your review, as well as your valuable suggestions on this paper. Your comments regarding inappropriate presentations were extremely valuable and helpful in revising and improving the paper. Thus, we have studied the comments carefully and have made the appropriate corrections. All inappropriate presentation has been modified and updated, and all amendments have been highlighted in red in the revised manuscript.

The problems you have suggested:

1. Page 4, Line 73

It is mentioned that “Unlike the global weak form, this method does not require the background..”

To my knowledge, the weak forms for meshfree, FEM, and XFEM are the same, and the differences are at the spatial discretization and interpolation approximation. It is recommended to double check whether you need a different weak form for LRPIM.

In addition, related references should be added since LRPIM has been quite popular.

Reply:

Thank you for your comments.Indeed, as you stated, the differences are at the spatial discretization and interpolation approximation, and the weak forms for meshfree, FEM, and XFEM are the same.

The difference is that LRPIM needs background grids to integrate at the spatial discretization and interpolation approximation, and I do not need a different weak form for LRPIM.

We have added 5 related references in line 419~421, line 454~461, line 494~497.

“[6] Rashadul Islam M , Vallejo M J , Tarefder R A . Crack Propagation in Hot Mix Asphalt Overlay Using Extended Finite-Element Model[J]. Journal of Materials in Civil Engineering, 2017, 29(5):04016296.

[17] Dehghan M, Haghjoo-Saniji M. The local radial point interpolation meshless method for solving Maxwell equations [J]. Engineering with Computers, 2017.

[18]Jianxing Mao,Dianyin Hu,Da Li,Rongqiao Wang,Jun Song. Novel adaptive surrogate model

based on LRPIM for probabilistic analysis of turbine disc [J]. Aerospace Science and Technology, 2017, 70.

[19] Jie Chen, Hai Wang, Guanghui Qing. Modeling vibration behavior of delaminated composite laminates using meshfree method in Hamilton system [J]. Applied Mathematics and Mechanics, 2015, 36(5).

[32] Nuismer R J . An energy release rate criterion for mixed mode fracture[J]. International Journal of Fracture, 1975, 11(2):245-250.”

2. Page 6, Line 108. Failure criterion used for XFEM was not indicated

It is also recommended to present the failure criterion used in XFEM modeling.

Reply:

We have fully absorbed your suggestions and present the failure criterion used in XFEM modeling in line 156~158 “The maximum circumferential tensile stress fracture criterion is built in ABAQUS, which realizes the expansion of cracks.”

3. Page 11, Line 193. A detailed description of Fig. 2 is needed

Since Fig. 2 has sub-plots of A,B,C,D, more detailed descriptions on those figures are needed.

Reply:

Thanks for your suggestions. We have fully absorbed your suggestions and Figure 2 is explained in more detail in line 209~213 “As shown in Fig.2, to propagate the fracture surface, the displacement field is calculated first, and then, the stress intensity factor and energy release rate of each crack front point are obtained according to the above method, including point a, b, c, d, e, f. Once a crack front point satisfies the crack propagation condition, such as b, c, e. The new positions of each crack front point are determined, like b',c',e'.”

4. Page 12, Line 211. The description on elastoplastic theory is incomplete and easily gets misunderstandings. Elastoplastic is generally used to quantify the stress-strain relations of materials with clear hardening or softening behaviors. Thus, a phenomenon-based yielding criterion and flow

rule are required to generate an elasto-plastic matrix. The authors presents a Newton-Raphson approach of solving non-linear equations instead of a elastoplastic theory. The title of 3.3 or the content should be rewritten to have a consistent description. By the side, it should be well documented that how to derive strains. Is there plastic strains? If only a Griffith type of failure criterion was applied, it seems that there is only elastic strains existing in non-fracture domain.

Reply:

Thanks for your suggestions and we have rename Section 3.3. Line 223 “Elastoplastic Calculation”.

We have further explained how to obtain strain in line 236~237 “

$$\boldsymbol{\varepsilon}_{(6 \times 1)} = \mathbf{B}_{(6 \times 3n)} \mathbf{u}_{(3n \times 1)} \quad (20)$$

$$\mathbf{B}_{(6 \times 3n)} = \begin{bmatrix} \frac{\partial \phi_1}{\partial x} & 0 & 0 & \dots & \frac{\partial \phi_n}{\partial x} & 0 & 0 \\ 0 & \frac{\partial \phi_1}{\partial y} & 0 & \dots & 0 & \frac{\partial \phi_n}{\partial y} & 0 \\ 0 & 0 & \frac{\partial \phi_1}{\partial z} & \dots & 0 & 0 & \frac{\partial \phi_n}{\partial z} \\ 0 & \frac{\partial \phi_1}{\partial z} & \frac{\partial \phi_1}{\partial y} & \dots & 0 & \frac{\partial \phi_n}{\partial z} & \frac{\partial \phi_n}{\partial y} \\ \frac{\partial \phi_1}{\partial z} & 0 & \frac{\partial \phi_1}{\partial x} & \dots & \frac{\partial \phi_n}{\partial z} & 0 & \frac{\partial \phi_1}{\partial x} \\ \frac{\partial \phi_1}{\partial y} & \frac{\partial \phi_1}{\partial x} & 0 & \dots & \frac{\partial \phi_n}{\partial y} & \frac{\partial \phi_1}{\partial x} & 0 \end{bmatrix} \quad (21)$$

Φ is shape function constructed by RPIM method.”

5. Some minor issues

1) Units for nephogram plots should be added to the captions. (e.g., Fig. 6)

2) Page 2, Line 8

“omparative” change to “Comparative”

3) Page 8, Line 136

4) “vectoral” change to “vector al”

5) Page 9, Line 149

“In this section, we briefly introduce the basic principles of the LRPIM and XFEM.” should be deleted or replaced by a relevant sentence.

6) Page 9, Line 161

References for fracture propagation theory should be added.

7) Page 10, Line 180

There are some symbols which do not appear correctly.

8) Page 11, Line 189

uc1, uc1, and uc3 should defined.

9) Page 14, Line 231

The sentence is not complete.

10) Page 21, Line 321

“mpa” change to “MPa”

11) Page 25, Line 360

Table2-Table 4 only give information of stress-displacement information instead of stress-strain information.

12) The format of references should be adjusted for this journal.

Reply:

Thank you for your valuable suggestions.

We have added units for nephogram plots, including Fig. 6, Fig. 7, Fig. 9, Fig. 10, Fig. 12, Fig. 13, Fig. 15, Fig. 16, Fig. 18, Fig. 19, Fig. 21, Fig. 22.

“vectoral” has been changed to “vector a_i ” in line 144.

The sentence “In this section, we briefly introduce the basic principles of the LRPIM and XFEM.” has been deleted.

We have added related references in line 419~421, line 454~461.

“[6] Rashadul Islam M , Vallejo M J , Tarefder R A . Crack Propagation in Hot Mix Asphalt Overlay Using Extended Finite-Element Model[J]. Journal of Materials in Civil Engineering, 2017, 29(5):04016296.

[17] Dehghan M, Haghjoo-Saniji M. The local radial point interpolation meshless method for solving Maxwell equations [J]. Engineering with Computers, 2017.

[18] Jianxing Mao, Dianyin Hu, Da Li, Rongqiao Wang, Jun Song. Novel adaptive surrogate model based on LRPIM for probabilistic analysis of turbine disc [J]. *Aerospace Science and Technology*, 2017, 70.

[19] Jie Chen, Hai Wang, Guanghui Qing. Modeling vibration behavior of delaminated composite laminates using meshfree method in Hamilton system [J]. *Applied Mathematics and Mechanics*, 2015, 36(5).

Symbols which do not appear correctly have been modified in line 187~188 “where θ_0 is the extension angle, or the angle between the extension direction and the x-axis, as shown in Figure 1 In the formula, when < 0 , take the "+" number, and when > 0 , take the "-" number”.

uc1, uc1, and uc3 have been defined in line 197~199 “ u_{c1} 、 u_{c2} 、 u_{c3} are the displacements of points in the local coordinate system, and $(\theta = +\pi)$ 、 $(\theta = -\pi)$ indicates whether the point is located on or below the fracture surface”.

That sentence has been reinterpreted in line 243~245 “The emphasis of this paper is to combine elastic-plastic calculation, crack propagation and meshless methods to solve model expansion. The crack propagation process is as follows:”.

“mpa” has been changed to “MPa” in line 329.

“stress-strain” has been replaced by “stress-displacement” in line 330.

We have checked the format of the references and revised some of them.

Reviewer 2

Thank you very much for your review, as well as your valuable suggestions on this paper. Your comments regarding inappropriate presentations were extremely valuable and helpful in revising and improving the paper. Thus, we have studied the comments carefully and have made the appropriate corrections. All inappropriate presentation has been modified and updated, and all amendments have been highlighted in red in the revised manuscript.

1. This paper presents so much theoretical literature. The actual work is very limited and confined to

the numerical analysis only. No validation is provided. Please mention this limitation in the discussion or conclusion section. Most of the references are more than 10 years old. In fact, more than 5-year old references is very often discouraged. Delete some old ref and include some new. One example, Modeling Crack Propagation in Asphalt Concrete Overlay using Extended Finite Element Model, ASCE Journal of Materials in Civil Engineering, 29(5). The flow of the paper is good, but, please write in passive format. Lines 149, 243: ‘We’?

Reply:

Thank you for your valuable suggestions. Indeed, as you stated, the actual work is very limited and confined to the numerical analysis only. We have mention this limitation in line 378~379 “The main work in this paper is numerical analysis, no validation is provided”.

We have added related references in line 419~421, line 454~461.

“[6] Rashadul Islam M , Vallejo M J , Tarefder R A . Crack Propagation in Hot Mix Asphalt Overlay Using Extended Finite-Element Model[J]. Journal of Materials in Civil Engineering, 2017, 29(5):04016296.

[17] Dehghan M, Haghjoo-Saniji M. The local radial point interpolation meshless method for solving Maxwell equations [J]. Engineering with Computers, 2017.

[18]Jianxing Mao,Dianyin Hu, Da Li,Rongqiao Wang,Jun Song. Novel adaptive surrogate model based on LRPIM for probabilistic analysis of turbine disc [J]. Aerospace Science and Technology, 2017, 70.

[19] Jie Chen,Hai Wang,Guanghui Qing. Modeling vibration behavior of delaminated composite laminates using meshfree method in Hamilton system [J]. Applied Mathematics and Mechanics, 2015, 36(5).

We have rewritten that sentence in line 243~245 “The emphasis of this paper is to combine elastic-plastic calculation, crack propagation and meshless methods to solve model expansion. The crack propagation process is as follows”.

Reviewer 3

Thank you very much for your review, as well as your valuable suggestions on this paper. Your comments regarding inappropriate presentations were extremely valuable and helpful in revising and improving the paper. Thus, we have studied the comments carefully and have made the appropriate corrections. All inappropriate presentation has been modified and updated, and all amendments have been highlighted in red in the revised manuscript.

The problems you have suggested:

1. line 175: "maximum circumferential normal stress" needs to be corrected as "maximum circumferential stress" or "maximum normal stress".

Reply:

We revised the sentence in line 182~184 “For mode I-II composite cracks, the direction of maximum energy release rate is the direction of the maximum circumferential stress. According to the theory of the maximum circumferential normal stress, the direction of propagation is as follows”.

2. line 181:incomplete sentence

Reply:

We revised the sentence in line 188 and delete “Critical energy release rate”.

3. Equation (17): K_{II} should be K_I

Reply:

We revised the Equation (17)

$$K_{IC} = \cos \frac{\theta_0}{2} \left[K_I \cos^2 \frac{\theta_0}{2} - \frac{3}{2} K_{II} \sin \theta_0 \right]$$

4. Section 3.3 talks about elastoplastic theory: what is the plasticity model used in the paper? There are several models and it is not clear which one was assumed to hold in the calculations.

Reply:

The first reviewer also made suggestions on this section. By synthesizing the views of two reviewers, we have renamed the title of Section 3.3 in Line 221 “Elastoplastic Calculation”.

5. line 271: K_1 should be E_1 ?

Reply:

We changed K_1 to E_1 in line 227.

6. lines 215 through 229 need to be reorganized. There are several badly written phrases that make the sequence 1 through 6 not understandable.

Reply:

Thanks for your suggestions. We have fully absorbed your suggestions and revised these sentences in line 226~240.

- 1) In the first trial calculation, the stiffness matrix K is obtained by taking the initial elastic constants E_1 and ν_1 .
- 2) The first approximate displacement U_1 is obtained under load R . Thus, ε_1 and stress σ_1 are obtained, as shown in Fig. 3.
- 3) The corresponding stress σ_1 of the actual curve relationship is solved by ε_1 , as shown in point O_1 .
- 4) Because the setting of E_1 and ν_1 is inaccurate, M_1 and N_1 are not on the curve. Therefore, according to ε_1 , the corresponding stress σ_1 is found from the functional relation $\sigma = f(\varepsilon)$ in the graph, as shown in point O_1 .

$$\boldsymbol{\varepsilon}_{(6 \times 1)} = \mathbf{B}_{(6 \times 3n)} \mathbf{u}_{(3n \times 1)} \quad (20)$$

$$\mathbf{B}_{(6 \times 3n)} = \begin{bmatrix} \frac{\partial \phi_1}{\partial x} & 0 & 0 & \dots & \frac{\partial \phi_n}{\partial x} & 0 & 0 \\ 0 & \frac{\partial \phi_1}{\partial y} & 0 & \dots & 0 & \frac{\partial \phi_n}{\partial y} & 0 \\ 0 & 0 & \frac{\partial \phi_1}{\partial z} & \dots & 0 & 0 & \frac{\partial \phi_n}{\partial z} \\ 0 & \frac{\partial \phi_1}{\partial z} & \frac{\partial \phi_1}{\partial y} & \dots & 0 & \frac{\partial \phi_n}{\partial z} & \frac{\partial \phi_n}{\partial y} \\ \frac{\partial \phi_1}{\partial z} & 0 & \frac{\partial \phi_1}{\partial x} & \dots & \frac{\partial \phi_n}{\partial z} & 0 & \frac{\partial \phi_1}{\partial x} \\ \frac{\partial \phi_1}{\partial y} & \frac{\partial \phi_1}{\partial x} & 0 & \dots & \frac{\partial \phi_n}{\partial y} & \frac{\partial \phi_1}{\partial x} & 0 \end{bmatrix} \quad (21)$$

ϕ is shape function constructed by RPIM method

- 5) The slope of secant $O\bar{O}_1$ is the secant modulus E_2 , and then, K_2 is obtained.
- 6) Repeat 2-5 to obtain $M_2, N_3, O_2, M_3, N_3, O_3$ and so on. They approach the real solution points M and N on the curve. When the displacement value of the last two iterations is close and the error is less than the allowable value, the calculation is complete.”

7. line 231: incomplete sentence

Reply:

Thanks for your suggestions and we have revised these sentences in line 242~244 “The emphasis of this paper is to combine elastic-plastic calculation, crack propagation and meshless methods to solve model expansion. The crack propagation process is as follows”.

8. line 245: what is "f"?

Reply:

Thanks for your suggestions and we have changed “f” to “ σ_b ”.

9. line 246: the analysis is static, why density is needed?

Reply:

In order to compare with physical experiments, material self-weight must be taken into account, so density parameters are used.

10. line 250: Should Figure 12 be Figure 4?

Reply:

I'm sorry, this is a mistake. We have corrected it.

11. line 266 : explain why $-1e-3\text{mm}$ was selected as the displacement?

Reply:

In order to compare the results of LRPIM with those of XFEM, the same initial boundary conditions need to be set. Therefore, we set a smaller value, which can also be set to other reasonable values.

12. line 267: incomplete sentence

Reply:

We have fully absorbed your suggestions and revised the sentence in line 277~279 "The initial vertical displacement in the middle of the upper surface is $-1e-3\text{mm}$, and the x-direction displacement and stress nephogram of the model are obtained".

13. lines 273-274: explain or rephrase "after several expansions, the fracture surface basically expands...".

Reply:

Thanks for your suggestions and we have rephrase the sentence in line 284~286 "After several steps, the fracture tip basically extends along the vertical direction, and the angle between the vertical direction and the fracture surface extension direction is 0 degrees, which is in line with the expectation."

Appendix B

Reviewer 3

Thank you very much for your review, as well as your valuable suggestions on this paper. Your comments regarding inappropriate presentations were extremely valuable and helpful in revising and improving the paper. Thus, we have studied the comments carefully and have made the appropriate corrections. All inappropriate presentation has been modified and updated, and all amendments have been highlighted in red in the revised manuscript.

The problems you have suggested:

4. Section 3.3 talks about elastoplastic theory: what is the plasticity model used in the paper? There are several models and it is not clear which one was assumed to hold in the calculations.

Reply:

The first reviewer also made suggestions on this section. By synthesizing the views of two reviewers, we have renamed the title of Section 3.3 in Line 221 “Elastoplastic Calculation”.

The plasticity model used in the paper is indicated in line 222~223 “The plasticity model was assumed to be elastic-perfectly model”

6. lines 215 through 229 need to be reorganized. There are several badly written phrases that make the sequence 1 through 6 not understandable.

Reply:

Thanks for your suggestions. We have fully absorbed your suggestions and revised these sentences once again in line 226~238.

- 1) In the first trial calculation, the stiffness matrix K is obtained by taking the initial elastic constants E_1 and ν_1 .
- 2) The first approximate displacement U_1 is obtained under load R . Thus, ϵ_1 and stress σ_1 are obtained, as shown in Fig. 3.
- 3) Because the setting of E_1 and ν_1 is inaccurate, M_1 and N_1 are not on the curve. Therefore, according to ϵ_1 , the corresponding stress σ'_1 is found from the functional relation $\sigma = f(\epsilon)$ in

the graph, as shown in point O_1 .

$$\boldsymbol{\varepsilon}_{(6 \times 1)} = \mathbf{B}_{(6 \times 3n)} \mathbf{u}_{(3n \times 1)} \quad (20)$$

$$\mathbf{B}_{(6 \times 3n)} = \begin{bmatrix} \frac{\partial \phi_1}{\partial x} & 0 & 0 & \dots & \frac{\partial \phi_n}{\partial x} & 0 & 0 \\ 0 & \frac{\partial \phi_1}{\partial y} & 0 & \dots & 0 & \frac{\partial \phi_n}{\partial y} & 0 \\ 0 & 0 & \frac{\partial \phi_1}{\partial z} & \dots & 0 & 0 & \frac{\partial \phi_n}{\partial z} \\ 0 & \frac{\partial \phi_1}{\partial z} & \frac{\partial \phi_1}{\partial y} & \dots & 0 & \frac{\partial \phi_n}{\partial z} & \frac{\partial \phi_n}{\partial y} \\ \frac{\partial \phi_1}{\partial z} & 0 & \frac{\partial \phi_1}{\partial x} & \dots & \frac{\partial \phi_n}{\partial z} & 0 & \frac{\partial \phi_1}{\partial x} \\ \frac{\partial \phi_1}{\partial y} & \frac{\partial \phi_1}{\partial x} & 0 & \dots & \frac{\partial \phi_n}{\partial y} & \frac{\partial \phi_1}{\partial x} & 0 \end{bmatrix} \quad (21)$$

Φ is shape function constructed by RPIM method

- 4) The slope of secant $\overline{OO_1}$ is the secant modulus E_2 , and then, K_2 is obtained.
- 5) Repeat 2-4 to obtain $M_2, N_3, O_2, M_3, N_3, O_3$ and so on. They approach the real solution points M and N on the curve. When the displacement value of the last two iterations is close and the error is less than the allowable value, the calculation is complete.”